

# Production of cupcake-like dessert containing microbial biosurfactant as an emulsifier

Ivison A. Silva[1,2], Bruno O. Veras[1], Beatriz G. Ribeiro[1],
Jaciana S. Aguiar[1], Jenyffer M. Campos Guerra[1], Juliana M. Luna[2,3] and
Leonie A. Sarubbo[2,3]

[1] Universidade Federal de Pernambuco, Recife, Pernambuco, Brazil
[2] Instituto Avançado de Tecnologia e Inovação (IATI), Recife, Pernambuco, Brazil
[3] Universidade Católica de Pernambuco, Recife, Pernambuco, Brazil

## ABSTRACT

This work describes the application of the biosurfactant from *Candida bombicola* URM 3718 as a meal additive like cupcake. The biosurfactant was produced in a culture medium containing 5% sugar cane molasses, 5% residual soybean oil and 3% corn steep liquor. The surface and interfacial tension of the biosurfactant were 30.790 ± 0.04 mN/m and 0.730 ± 0.05 mN/m, respectively. The yield in isolated biosurfactant was 25 ± 1.02 g/L and the CMC was 0.5 g/L. The emulsions of the isolated biosurfactant with vegetable oils showed satisfactory results. The microphotographs of the emulsions showed that increasing the concentration of biosurfactant decreased the oil droplets, increasing the stability of the emulsions. The biosurfactant was incorporated into the cupcake dessert formulation, replacing 50%, 75% and 100% of the vegetable fat in the standard formulation. Thermal analysis showed that the biosurfactant is stable for cooking cupcakes (180 °C). The biosurfactant proved to be promising for application in foods low in antioxidants and did not show cytotoxic potential in the tested cell lines. Cupcakes with biosurfactant incorporated in their dough did not show significant differences in physical and physical–chemical properties after baking when compared to the standard formulation. In this way, the biosurfactant has potential for application in the food industry as an emulsifier for flour dessert.

## INTRODUCTION

In the era of globalization, biotechnology faces the challenge of producing biocompatible compounds that meet the needs of the current market (*Kieliszek et al., 2017*). In this context, the amphipathic natural additives produced by microorganisms and known as biosurfactants (*Campos, Stamford & Sarubbo, 2014*) stand out.

The use of biosurfactants in the food industry is due to their emulsifying, foaming, humectant and solubilizing properties; that is, these biomolecules can be used as emulsifiers in the processing of raw materials, in the control of the agglomeration of fat

Corresponding author
Leonie A. Sarubbo,
leonie.sarubbo@unicap.br

globules, in the stabilization of aerated systems and to improve the consistency of fat products (*Salek & Euston, 2019*).

The industrial use of microbial biosurfactants, however, has been hampered due to the high production costs associated with inefficient recovery methods and the use of expensive substrates. However, these costs can be significantly reduced by using alternative sources of nutrients, as well as by obtaining high product yields and applying more economical extraction methods (*Salek & Euston, 2019*). On the other hand, the food industry still does not use biosurfactants as additives on a large scale, as the regulation for approval of these biomolecules as a new food ingredient needs approval, which requires time and industrial investment.

Due to changes in the behavior of eating habits, ready-to-eat foods, often available as small units and known as "snackers", are becoming increasingly important. Examples of these products include donuts, muffins or cupcakes and cookies (*Zouari et al., 2016*).

Cupcakes are characterized by a long shelf life and can undergo changes in their composition in order to provide special dietary needs (*Gupta, Bawa & Abu-Ghannam, 2011*). Another important aspect in the offering of desserts is its nutritional value, considering that consumer satisfaction is decisive for the success of a newly formulated product (*Skrbic & Cvejanov, 2011*).

Trends in healthy eating and growing progress in research into functional natural food additives have been noted, including those that exhibit antioxidant activity without side effects compared to synthetic compounds (*Abdel-Mawgoud & Stephanopoulos, 2018*; *Bandyopadhyay, Chakraborty & Bhattacharyya, 2014*). Thus, in order to reduce the use of synthetic emulsifiers or constituents of low nutritional value in foods, the use of biosurfactants has aroused industrial interest (*Campos et al., 2013*).

Among the micro-organisms that produce bioemulsifiers, yeasts of the genus *Candida* have been suggested due to their beneficial use in food. A major advantage of using yeasts is their Generally Regarded as Safe (GRAS) status, which classifies them as safe because they do not present risks of toxicity and pathogenicity (*Campos, Stamford & Sarubbo, 2014*).

In this sense, the objective of this work was to evaluate the addition of a biosurfactant with an emulsifying capacity to replace the vegetable fat used in the formulation of cupcakes with a view to providing a more biocompatible product for the food industries.

## MATERIALS AND METHODS

### Microorganism

The yeast *C. bombicola* URM 3718, deposited in the Culture Collection of the Department of Mycology from Federal University of Pernambuco, was used in the production of the biosurfactant. Yeast maintenance was performed using Yeast Mold Agar (YMA) containing yeast extract (0.3%), D-glucose (1%), peptone (0.5%) and agar (2%), pH 7.0. The growth medium, Yeast Mold Broth (YMB), had the same composition, excluding agar.

## Biosurfactant production

The yeast inoculum was standardized by transferring a loopful of the cream colored young culture to flasks containing 50 mL of YMB medium and incubated at 150 rpm at 28 °C for 24 h. After this period, dilutions were performed until the final concentration of $10^4$ cells/mL was obtained, which was used in the concentration of 5% (v/v). The biosurfactant was produced in a medium formulated with distilled water containing 5% sugar cane molasses, 5% residual soybean frying oil and 3% corn steep liquor (pH 6.0), as described by *Freitas et al. (2016)*. The fermentations were carried out in 1,000 mL Erlenmeyer flasks, containing 500 mL of the production medium and incubated with the inoculum, under orbital shaking at 180 rpm for 120 h at 28 °C.

## Determination of surface and interfacial tension

The surface tension of the biosurfactant was measured in the cell-free broth obtained after centrifugation at 5,000 rpm for 20 min, in a KSV Sigma 700 (Helsinki, Finland) tensiometer employing the Du Nouy ring method at room temperature. The interfacial tension was measured in the same way in relation to *n*-hexadecane.

## Isolation of biosurfactant

A method developed in the laboratory was used, which initially consisted of extraction with ethyl acetate, twice, in the proportion 1:4 (v/v) of the non-centrifuged broth. Then, the organic phase was centrifuged ($2,600 \times g$ for 20 min) and filtered. The filtrate was transferred back to the separating funnel and a saturated NaCl solution was added to separate the remaining aqueous phase. The organic phase was transferred to an Erlenmeyer flask and anhydrous $MgSO_4$ was added until granules were formed, which were filtered and dried at 50 °C.

## Composition of biosurfactant

The biochemical composition of the isolated biosurfactant was determined as described by *Luna et al. (2016)*. The protein concentration in the isolated biosurfactant was determined with bovine albumin as the standard. Total carbohydrate content was determined using the phenol–sulphuric acid method and lipids were quantified from extraction with chloroform: methanol at different proportions (1:1 and 1:2, v/v).

## Critical micelle concentration

For this determination, NaOH diluted in distilled water was added to the crude extract of the biosurfactant, in the proportion 1:7 (v/v). Then, the product formed was washed with acetone, filtered and dried to evaporate the solvent. To obtain the CMC, 0.1 g of the product was weighed and successive dilutions with distilled water (concentrations of biosurfactant from 0 to 700 ppm) were performed and the surface tensions were determined in tensiometer (Sigma 700; KSV Instruments, Helsinki, Finland) using the NUOY ring up to a constant value (standard deviation less than 0.4 mN/m during 10 successive measurements). The CMC was obtained by plotting surface tension against surfactant concentration and expressed as g/L of biosurfactant.

## Emulsification activity

The emulsification activity of the biosurfactant was compared with guar gum using the method described by *Prasanna, Bell & Grandison (2012)*. Two mL of a vegetable oil (corn oil, soybean oil, sunflower oil, canola oil and peanut oil) were added to two mL of a solution of the biosurfactant at half the CMC (1/2CMC), at the CMC and twice the CMC (2 × CMC) or solution of guar gum (1%, w/v) in a glass tube with screw cap and the contents were vortexed for 2 min at 50 Hz. After 24 h, the emulsification index ($E_{24}$) was determined according to Eq. (1):

$$E_{24} = (h_e/h_t) \times 100 \tag{1}$$

where: $h_e$ is the height of the emulsion layer and $h_t$ is the total height of the mixture, in mm. All samples were stored at 27 °C (*Han et al., 2015*).

## Light microscopy of emulsions

The particle size distribution of the emulsions was measured using a modification of the method described by *Prasanna, Bell & Grandison (2012)*. Briefly, an optical microscope (XSZ-HS3; Zhongyi Ltd., Beijing, China) was used to examine and photograph the emulsion after 24 h of storage at 27 °C through a 10× objective lens. A volume of 60 μL of the emulsion was added to a slide and left to stand for 5 min for stabilization of the emulsion, followed by observation under the microscope (*Han et al., 2015*).

## Differential scanning calorimetry and thermogravimetric analysis

The thermal analysis of the biosurfactant (50 mg) was performed according to *Han et al. (2015)*, using a simultaneous thermal analyzer STA 449 F3 (NETZSCH, Selb, Germany). For this, successive heating/cooling/heating steps were performed, with a heating and cooling ratio of 10 °C/min, in a nitrogen atmosphere with a flow rate of 50.0 mL/min, in the range of 40 at 400 °C.

## Antioxidant activity

### Evaluation of the sequestering activity of the radical 2,2′-azino-bis (3-ethyl-benzothiazoline-6-sulfonate) (ABTS$^{\bullet+}$)

This analysis was carried out following the methodology described by *Re et al. (1999)*. The ABTS$^{\bullet+}$ (Sigma–Aldrich, Dorset, UK) radical was formed by the reaction of five mL of the ABTS$^{\bullet+}$ 7 mM solution, with 88 μL of the 140 mM potassium persulfate solution, incubated at 25 °C and in the absence of light for 16 h. Once formed, the radical was diluted with P.A. ethanol until the absorbance value of 0.700 ± 0.020 at 734 nm was obtained. Different concentrations (156.25, 312.50, 625.00, 1,250.00, 2,500.00 and 5,000.00 μg/mL) of the biosurfactant (10 μL) were mixed with ABTS$^{\bullet+}$ solution (one mL) and after 6 min incubation in the dark, the absorbances were read. Measurements were performed in triplicates and inhibition activities were calculated based on the percentage of ABTS$^{\bullet+}$ sequestered. For the calibration curve, a standard solution of Trolox (6-hydroxy-2,5,7,8-tetramethychroman-2-carboxylic acid; Aldrich Chemical Co., Dorset,

UK), a synthetic antioxidant analogous to vitamin E, was prepared at a concentration of 100 to 2,000 μM. The percentage of inhibition (I%) was calculated using Eq. (2):

$$\%I = [(Abs_0 - Abs_1)/Abs_0] \times 100 \tag{2}$$

where $Abs_0$ is the absorbance of the control and $Abs_1$ is the absorbance of the compound.

### Evaluation of the sequestering activity of the 2,2-diphenyl-1-picrylhydrazyl radical

The evaluation of the antioxidant activity of the biosurfactant by the free radical scavenging method was measured by means of hydrogen donation using the stable radical DPPH (*Blois, 1958*). A stock solution of methanolic DPPH (200 μM) was further diluted in methanol to reach an UV–VIS absorbance between 0.6 and 0.7 at 517 nm, obtaining the DPPH working solution. Different concentrations of the composition (40 μL) were mixed with DPPH solution (250 μL) and after 30 min of incubation in the dark, the absorbances were read at the same wavelength mentioned above. The measurements were carried out in triplicates and the inhibition activities were calculated based on the percentage of DPPH eliminated. The percentage of inhibition (I%) was calculated using Eq. (3):

$$I\% = [(Abs_0 - Abs_1)/Abs_0] \times 100 \tag{3}$$

where $Abs_0$ is the absorbance of the control and $Abs_1$ is the absorbance of the compound.

For the calibration curve, a standard Trolox solution was prepared, at a concentration of 10–200 μM.

### Total antioxidant capacity by phosphomolybdenum

The total antioxidant capacity (TAC) of the biosurfactant was evaluated by the phosphomolybdenum method, which consisted of the composition's ability to reduce molybdenum and form the phosphate-molybdate complex (*Pietro, Pineda & Aguilar, 1999*). Different concentrations of the composition (100 μL) were mixed in one mL with phosphate-molybdate complex (40 mM ammonium molybdate/60 mM sulfuric acid, 280 mM sodium phosphate), then incubated in a water bath at 90 °C for 90 min. The absorbances were read on a spectrophotometer at 695 nm. For calculation purposes, ascorbic acid was considered to has 100% antioxidant activity. The activity was calculated according to Eq. (4):

$$\%\text{Active antioxidant} = [(Abs_1 - Abs_0)/Abs_0 - Abs_{AA}] \times 100 \tag{4}$$

where $Abs_0$ is the absorbance of the control, $Abs_1$ is the absorbance of the compound and $Abs_{AA}$ is the absorbance of ascorbic acid.

## Evaluation of the cytotoxic potential of the biosurfactant

The MTT ((3-(4,5-dimethylthiazol-2-yl)-2,5-diphenyltetrazolium bromide) assay was carried out with mice fibroblast cells (L929) and renal epithelial cells of African green

**Table 1 Cupcake dough using standard formulation (100% vegetable fat) and formulations A (50% vegetable fat + 50% biosurfactant), B (25% vegetable fat + 75% biosurfactant) and C (100% biosurfactant).**

| Ingredients | Doughs | | | |
|---|---|---|---|---|
| | Standard (g) | Formulation A (g) | Formulation B (g) | Formulation C (g) |
| Wheat flour | 153.66 | 153.66 | 153.66 | 153.66 |
| Sugar | 92.20 | 92.20 | 92.20 | 92.20 |
| Baking powder | 7.66 | 7.66 | 7.66 | 7.66 |
| Salt | 1.80 | 1.80 | 1.80 | 1.80 |
| Skim milk powder | 11.60 | 11.60 | 11.60 | 11.60 |
| Vegetable fat | 77.00 | 38.50 | 19.25 | 0.00 |
| Liquid whole egg | 46.10 | 46.10 | 46.10 | 46.10 |
| water | 76.80 | 76.80 | 76.80 | 76.80 |
| Biosurfactant | 0.00 | 38.50 | 57.75 | 77.00 |
| Total | 466.82 | 466.82 | 466.82 | 466.82 |

monkey (Vero), cell obtained from the Rio de Janeiro cell bank (Rio de Janeiro, Brazil), as described by *Resende et al. (2019)*. The cells were detached with trypsin solution (0.5%) and added at a concentration of $10^5$/mL to DMEM medium in a 96-well microplate, followed by incubation at 37 °C in a 5% $CO_2$ atmosphere for 24 h. Next, 10 µL of the isolated biosurfactant solutions at concentration of 6.25, 12.50, 25.00 and 50.00 µg/mL were added, followed by incubation under the same atmospheric conditions for 72 h. The negative control was DMEM and the positive control was phosphate buffer (150 mmol/L, pH 7.4). After 72 h, 25 µL of MTT (five mg/mL) stain was added and incubated for 3 h. After incubation, the culture medium with MTT was aspirated and 100 µl of dimethyl sulphoxide were added for the spectrophotometric reading at 560 nm. The percentage inhibition was calculated using GraphPad Prism 7.0 demo software.

## Dough preparation and cupcakes baking

Commercial samples of the following ingredients were purchased for processing the cupcakes: white wheat flour, vegetable fat (margarine) made from vegetable oils (palm and soy) and enriched with vitamins A and E (80% fat and 16% moisture), refined sugar, fresh eggs, baking powder (containing sodium pyrophosphate, sodium bicarbonate and corn starch), skim milk powder and salt.

A cake dough formulation was used as a reference (*Cross, 2006*), according to Table 1. Vegetable fat and liquid whole egg were beaten with a N50 planetary mixer (Hobart GmbH, Offenburg, Germany) for 60 s (speed 3). The dry ingredients, weighed in a bowl and mixed with water were added to the pre-beaten vegetable fat/egg mass in the mixing unit and beat for more 60 s. After filling the dough into paper forms in 50 ± 0.2 g aliquots, the cupcakes were baked in a MIWE oven (Michael Wenz GmbH, Arnstein, Germany) for 24 min at 180 °C. After cooling, the cupcakes were stored in plastic bags until analysis (18–24 h after cooking). From the reference formulation, 50%

(formulation A), 75% (formulation B) and 100% (formulation C) of vegetable fat was replaced by the biosurfactant. The cupcake formulations were processed in triplicate.

## Physical properties of cupcakes after baking

Weight, diameter and height were evaluated. Weight was determined using a 0.001 g precision analytical balance (BEL Engineering, Monza, Italy). The diameter of the cupcakes was measured according to *Noor Aziah, Mohamad Noor & Ho (2012)*. Four samples selected randomly were put next to one another and the total diameter was measured with a caliper (Mitiyoto Co., Tokyo, Japan). Then, all cupcakes were rotated by 90 °C and the new diameter was measured. The average of the two measurements divided by four was taken as the final diameter. The height of the cupcakes was also measured using a Digit CAL SI caliper (Mtx, Winslow, IL, USA).

## Physical–chemical analysis of cupcakes after baking

Test methodologies were used according to *AOAC (2002)*. The moisture content was determined using the gravimetric method, based on the weight loss of the samples submitted to heating in an oven at 105 °C until constant weight. The total protein concentration was calculated using the Kjeldahl method, based on the acid digestion of organic matter followed by distillation, with nitrogen subsequently dosed by titration; and the nitrogen value multiplied by the 6.25 factor. For the fixed mineral residue (ashes) the gravimetric method was used, based on the determination of the weight loss of the samples submitted to incineration at 550 °C.

To quantify the lipid fraction, the Bligh–Dyer cold extraction method (*Bligh & Dyer, 1959*) was used, in which a mixture of chloroform, methanol and water is used. The carbohydrate content was obtained from the difference between 100 and the sum of the moisture, protein, lipid and ash determinations.

## Determination of the energy value of cupcakes after baking

To determine the energy value, the mass values of carbohydrates, lipids and proteins were multiplied by 4, 9 and 4, respectively (*Pires et al., 2017*).

## Statistical analysis

All analyzes were performed in triplicate. Means and standard errors were calculated using Microsoft Office Excel 2016. The Tukey Test, with 95% confidence, was used in the physical–chemical analysis of formulated cupcakes.

# RESULTS

## Surface and interfacial tension

The most important properties for verifying the effectiveness of a biosurfactant are related to surface and interfacial tensions. Biosurfactants work by reducing the forces that exist between molecules on the liquid's surface, exerting influence on hydrogen bonds and even on hydrophobic–hydrophilic interactions, thus increasing the contact surface (*Bezerra et al., 2018*). The results obtained for surface and interfacial tension for

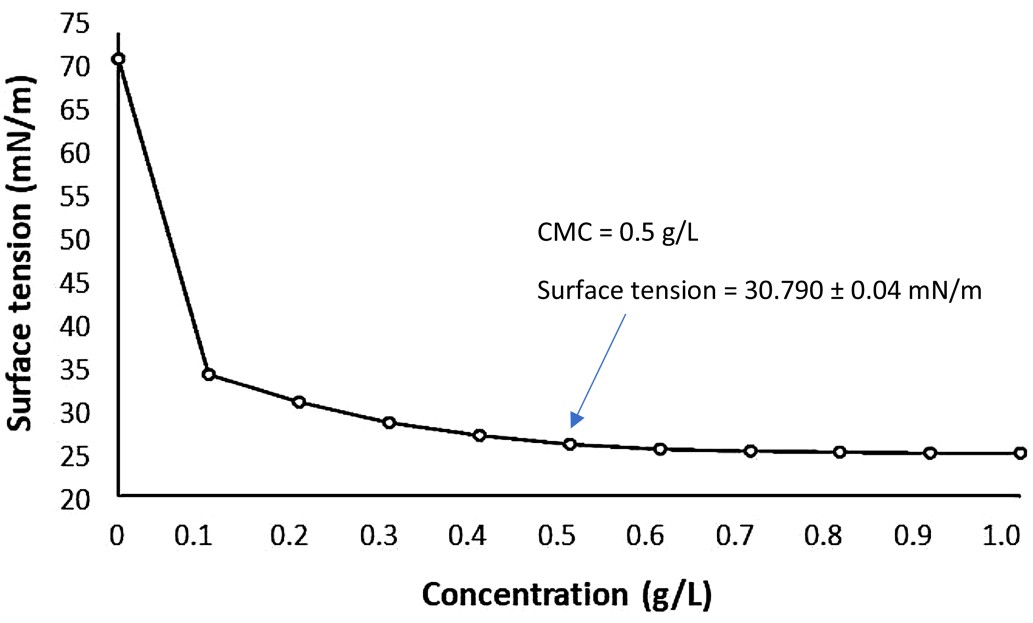

**Figure 1 Critical Micelle Concentration (CMC) of the biosurfactant isolated from *Candida bombicola* URM 3718.**

the biosurfactant produced by *C. bombicola* were 30.790 ± 0.04 mN/m and 0.730 ± 0.05 mN/m, respectively. According to *Akbari et al. (2018)*, biosurfactants with the ability to reduce the surface tension of water from 72 to 35 mN/m and the interfacial tension from 40 to 1 mN/m are effective, being considered good surfactants.

## Isolation and composition of biosurfactant

The biosurfactant was isolated from a procedure developed in the laboratory. The method has advantages over other methods conventionally used for the extraction of biosurfactants, since initial stages of centrifugation and filtration are discarded, in addition to the use of a lower volume of solvent (ethyl acetate). The biosurfactant yield obtained was 25.00 ± 1.02 g/L. The determination of the biochemical composition of the biosurfactant revealed the presence of 70% lipids and 30% carbohydrates.

## Critical micelle concentration

CMC corresponds to the lowest concentration of surfactant necessary to reduce surface tension as much as possible. This condition is reached when micelles start to form in solution and there is no further reduction in surface tension (*Rocha e Silva et al., 2018*).

The reduction in surface tension as a function of the concentration of biosurfactant is shown in Fig. 1. The CMC of the biosurfactant was determined as 0.5 g/L. CMC is related to the efficiency of surfactants and ranges from 0.001 to 2 g/L for biosurfactants. Biosurfactants are more efficient as they have reduced CMC (*Santos et al., 2016*). Therefore, we can say that the CMC of the biosurfactant from *C. bombicola* is within the acceptable range for biosurfactants.
**Table 2 Emulsification index (E$_{24}$) of guar gum and *C. bombicola* URM 3718 biosurfactant with vegetable oils.** Data expressed as mean ± S.D. of triplicate determinations.

| Guar gum (1%, p/v) | E$_{24}$ (%) | | | | |
|---|---|---|---|---|---|
| | Soybean oil | Corn oil | Canola oil | Sunflower oil | Peanut oil |
| | 48.11 ± 1.33 | 66.77 ± 0.15 | 52.59 ± 1.05 | 30.49 ± 0.85 | 42.36 ± 1.38 |
| **Biosurfactant concentration** | E$_{24}$ (%) | | | | |
| 1/2×CMC | 41.16 ± 0.32 | 39.29 ± 0.00 | 44.64 ± 0.27 | 49.78 ± 0.31 | 62.70 ± 0.89 |
| CMC | 45.81 ± 3.50 | 48.28 ± 0.00 | 45.44 ± 1.40 | 54.61 ± 0.79 | 68.97 ± 0.00 |
| 2×CMC | 51.78 ± 0.08 | 56.33 ± 0.48 | 50.86 ± 1.22 | 56.72 ± 2.19 | 69.48 ± 0.73 |

## Emulsification index (E$_{24}$)

The emulsification index (E$_{24}$) is a qualitative and fast method for evaluating the emulsifying properties of surfactants. Emulsion systems with hydrocarbons and/or water insoluble compounds are stabilized by surfactants. According to *Campos et al. (2015)*, the stability of these emulsions is indicative of the activity of biosurfactants, although the emulsification capacity is not related to the ability to reduce surface tension.

The emulsifying capacity of the *C. bombicola* biosurfactant with different vegetable oils was compared with guar gum and the results are shown in Table 2. Guar gum was used as a comparison, since it has regulation documented by the FDA as an emulsifier for food purposes (*Sharma et al., 2018*). The results obtained indicate that both bioemulsifiers were able to satisfactorily emulsify the studied oils. The increase in the concentration of biosurfactant, as expected, increased the emulsification percentages.

## Particle size distribution of emulsions

It is important that the emulsions formed in any process remain stable for marketing purposes. In this sense, the evaluation of the emulsion particle size distribution is necessary for the study of the physical properties of a biosurfactant, since it is possible to observe the emulsion droplet size (*Luo et al., 2017*; *Han et al., 2015*). The photomicrographs of the emulsions of the produced biosurfactant solutions (½ CMC, CMC and 2× CMC) with vegetable oils are illustrated in Fig. 2.

According to *Kokal (2005)*, both the size of the droplets and the form of their distribution depends on related factors, including surface tension, nature of the emulsifier, properties of the oil used, in addition to the shear rate (mixtures, etc.). Analyzing Fig. 2, it is observed, in general, that the increase in the concentration of biosurfactant was determinant for the decrease of the droplets, which is related to the stability of the emulsion. It is also possible to observe that the microphotographs for peanut oil showed a more uniform aspect regarding the size of the droplets, corroborating the results presented in Table 2, where the highest emulsification indexes were obtained.

## Differential scanning calorimetry and thermogravimetry

The thermostability profile of the produced biosurfactant is shown in Fig. 3. Regarding the thermogravimetric analysis, it was possible to obtain the mass loss curve of the

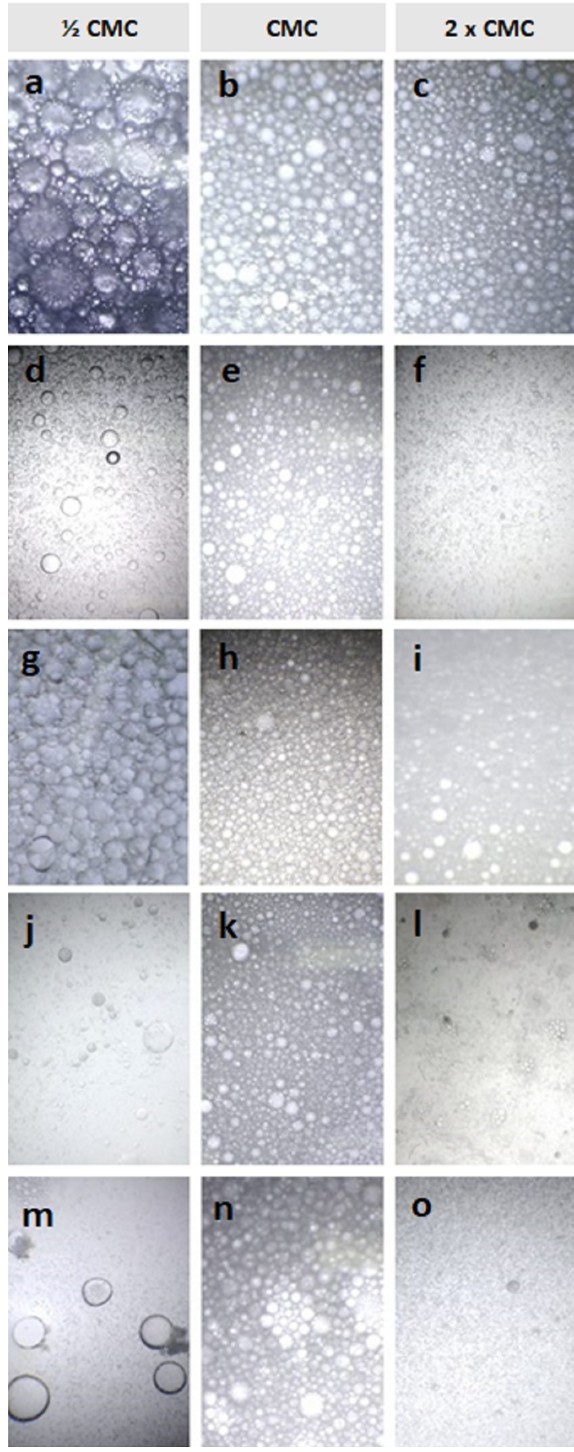

**Figure 2** Photomicrograph of the emulsions formed by the biosurfactant in different concentrations for the following vegetables oils: (A–C) peanut; (D–F) canola; (G–I) soybean; (J–L) sunflower and (M–O) corn.

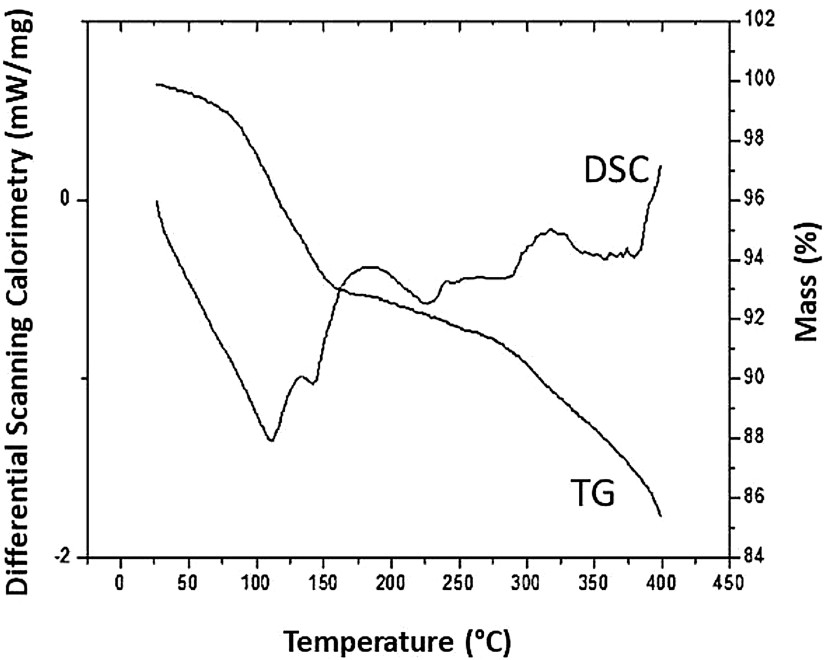

**Figure 3 DSC and TG of the *C. bombicola* URM 3718 biosurfactant.**

biosurfactant as a function of temperature. According to Fig. 3, the biosurfactant showed good thermal stability, with a loss of mass of 7.22% at a temperature of 181.76 °C. Up to the final temperature (400 °C), the variation in mass loss was in the range of 14.36%.

For DSC analysis, the biosurfactant thermogram showed an exothermic peak with a crystallization temperature of 116.76 °C (start temperature of 26.76 °C) and the endothermic melting peaks were observed at 186.76 °C (start temperature at 141.76 °C) and 319.26 °C (start temperature at 291.76 °C).

## Antioxidant activity

Lipid oxidation and enzymatic activities are problems of great significance in the food industry, which can result in changes in the chemical composition of the food, thus reducing its quality and shelf life. Antioxidants play a central role in neutralizing these processes, and their action can be verified through various mechanisms, that is, inhibiting free radicals, sequestering oxygen or chelating metal ions that catalyze oxidative reactions (*Cörmert & Gökmen, 2018*).

Catches of organic radicals such as DPPH or reduction of complexes such as phosphomolybdenum, were determined and the percentages of total antioxidant capacity of the solutions of the biosurfactant are shown in Table 3. The biosurfactant showed promising results in relation to the technique of reduction of the phosphomolybdenum complex. Comparing the percentages of the total antioxidant capacity (TAC) of the biosurfactant with the reference concentration of ascorbic acid (2,000 μg/mL), the biosurfactant showed 25.47% of activity. The results showed a linear relationship, indicating that the increase in the concentration of biosurfactant favors the increase in its

**Table 3 Percentage of DPPH radical scavenging (% I) and total antioxidant capacity (% TAC) at different concentrations of the biosurfactant.** Data expressed as mean ± S.D. of triplicate determinations.

| Biosurfactant concentration (µg/mL) | % I | % CAT |
| --- | --- | --- |
| 2,000.00 | – | 25.47 ± 3.18 |
| 1,000.00 | 52.42 ± 0.69 | 11.16 ± 0.37 |
| 500.00 | 32.93 ± 0.59 | 7.65 ± 4.53 |
| 250.00 | 25.05 ± 0.67 | 7.01 ± 0.46 |
| 125.00 | 8.48 ± 1.16 | 3.47 ± 0.07 |
| 62.50 | 4.55 ± 0.35 | 2.80 ± 0.21 |
| 31.25 | 2.32 ± 0.90 | – |

**Table 4 Percentage of sequestration (% I) of the radical 2,2′-azino-bis (3-ethylbenzothiazoline-6-sulfonate) (ABTS$^{\bullet+}$) by the biosurfactant.** Data expressed as mean ± S.D. of triplicate determinations.

| Biosurfactant concentration (µg/mL) | % I |
| --- | --- |
| 5,000.00 | 24.27 ± 2.38 |
| 2,500.00 | 17.40 ± 0.49 |
| 1,250.00 | 16.64 ± 0.44 |
| 625.00 | 3.36 ± 0.71 |
| 312.50 | 3.82 ± 0.35 |
| 156.25 | 0.61 ± 2.73 |

activity. Thus, in higher concentrations, the biosurfactant produced has the potential for application in foods lacking in antioxidants.

From the DPPH test, the ability of the biosurfactant to prevent oxidation of the DPPH radical was evaluated. Analyzing Table 3, it is possible to observe again a linear relationship with the reduction percentages of the DPPH radical (%I). Comparing the percentages with the Trolox reference, the biosurfactant showed 52.42% at a concentration of 1,000 µg/mL, demonstrating the potential for reducing the DPPH radical.

The third methodology used to determine the antioxidant profile of the biosurfactant was the method of sequestering the radical cation ABTS$^{\bullet+}$ (Table 4). The biosurfactant did not present good percentages of inhibition of the radical in comparison with the standard Trolox, which was able to inhibit in 84.58%, in the same concentration. Therefore, this test does not indicate the direct application of the biosurfactant as a sequester of the ABTS$^{\bullet+}$ radical cation.

## Cytotoxic evaluation

Several microbial metabolites often have an adverse effect on host organisms, triggering epidemic diseases, including neurotoxic and cytotoxic effects (Patowary et al., 2017). Therefore, a cytotoxic evaluation of these metabolites is extremely important.

The results of percentage of cell viability obtained for the analyzed concentrations of the biosurfactant are described in Table 5. It was observed that for both cell lines, the

**Table 5 Percentage of viability of the *C. bombicola* URM 3718 biosurfactant against the L929 and the Vero strains.**

| Concentrations (µg/mL) | Celular viability (%) | Satandard deviation (%) |
|---|---|---|
| L929 | | |
| Control | 99.50 | 3.07 |
| 6.25 | 109.79 | 1.56 |
| 12.50 | 92.06 | 6.56 |
| 25.00 | 93.14 | 10.52 |
| 50.00 | 107.66 | 5.00 |
| Vero | | |
| Control | 99.82 | 1.96 |
| 6.25 | 94.72 | 3.32 |
| 12.50 | 96.54 | 3.43 |
| 25.00 | 98.37 | 4.79 |
| 50.00 | 95.93 | 0.72 |

biosurfactant showed cell viability above 90%. According to *Gomes Silva et al. (2017)*, inhibition percentages from 1% to 20% are considered to be without inhibitory activity. Thus, the biosurfactant did not show cytotoxic potential compared to the studied strains. ISO 10993-5, from 2009, determines that a cell viability above 80% can be considered nontoxic in nature (*International Organization for Standardization, 2009*).

For the L929 strain, there were percentages of cell viability above 100%. For MTT testing especially this type of phenomenon is not uncommon. One reason for this may be the random experimental fluctuation, which should be between ±10%; another reason is stimulation by treatment. In the MTT assay, cell respiration is measured and its induction may indicate severe cell stress, since MTT is prone to compounds that interfere with energy metabolism, which can increase MTT metabolism to up to 200% of the base. A third reason, although almost disposable, would be the direct chemical reduction of MTT by the biosurfactant; however, when this phenomenon occurs, the values are excessively high (>500%).

## Characterization of cupcakes

The visual appearance of the cupcakes after baking is illustrated in Fig. 4, while the physical properties are described in Table 6.

The results demonstrate that there were no major discrepancies between the different formulations in relation to the evaluated parameters. Thus, the partial or total replacement of vegetable fat by the biosurfactant did not cause significant changes in the physical properties of the cupcakes, despite the more pronounced fluctuation observed in the values found for Height, which remained within an acceptable margin. Visually (Fig. 4) there were also no major differences in cupcakes, which remained similar and attractive. Making a comparison with the values found in Height and Weight of the standard formulation versus formulations with the addition of biosurfactant, it is possible to observe that even noting a variation in the Height values, the respective weight measurements did not

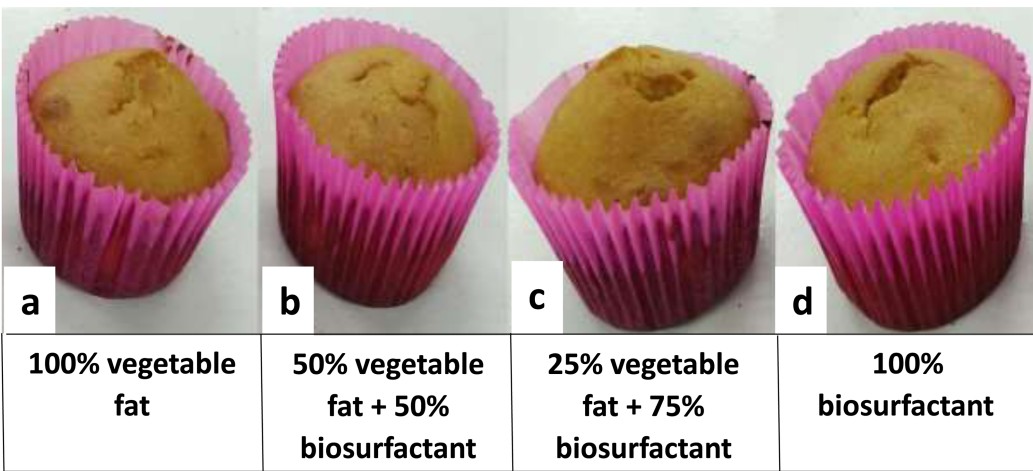

| a | b | c | d |
|---|---|---|---|
| 100% vegetable fat | 50% vegetable fat + 50% biosurfactant | 25% vegetable fat + 75% biosurfactant | 100% biosurfactant |

**Figure 4 Cupcakes after baking.** (A) Standard formulation; (B) formulation A; (C) formulation B and (D) formulation C.

**Table 6 Physical properties of cupcakes after baking for the formulations tested.** Data expressed as mean ± S.D. of triplicate determinations.

| Parameter | Standard formulation (100% vegetable fat) | Formulation A (50% vegetable fat + 50% biosurfactant) | Formulation B (25% vegetable fat + 75% biosurfactant) | Formulation C (100% biosurfactant) |
|---|---|---|---|---|
| Weight (g) | 44.841 ± 0.05 | 44.955 ± 0.22 | 43.712 ± 0.01 | 44.576 ± 0.29 |
| Height (mm) | 41.265 ± 1.18 | 35.975 ± 1.93 | 45.885 ± 0.94 | 42.435 ± 0.35 |
| Diameter (mm) | 59.120 ± 0.25 | 57.630 ± 0.01 | 59.400 ± 0.36 | 57.610 ± 0.68 |

undergo significant changes. This was due to the fact that the reduction of fat used during the preparation of cupcakes decreases the amount of air bubbles that are incorporated into the dough and expand during cooking, resulting in an increase in the density of crumbs and consequently, in an increase in dough (*Zahn, Pepke & Rohm, 2010*).

The results obtained for the physical–chemical composition and the energy value of the cupcakes are shown in Table 7. With respect to moisture, a significant decrease was observed between the three formulations proposed, with no significant difference between the standard formulations and formulation A. The vegetable fat used in the formulation of the cupcakes in this work, as an emulsion, is composed of an oily phase and an aqueous phase. As a consequence, cupcakes containing vegetable fat in their composition had a higher moisture content compared to cupcakes that contained biosurfactant in the formulation. Moisture can affect the shelf life of cupcakes, considering that it is related to the water activity present in the food, which favors microbial contamination.

There was no significant difference between the formulations regarding the levels of lipids found in the cupcake samples. Considering that the *C. bombicola* biosurfactant has a glycolipidic nature, all formulations presented a similar lipid content. However, it is possible to observe an increase in the percentage of lipids with the increase in the

**Table 7 Physical–chemical composition and energy value of cupcakes for the tested formulations.**
Data expressed as mean ± S.D. of triplicate determinations. Means in the same line with different letters are significantly different ($p \leq 0.05$) according to the Tukey Test.

| Parameter | Standard formulation (100% vegetable fat) | Formulation A (50% vegetable fat + 50% biosurfactant) | Formulation B (25% vegetable fat + 75% biosurfactant) | Formulation C (100% biosurfactant) |
|---|---|---|---|---|
| Moisture (%) | 26.43 ± 0.28[a] | 26.25 ± 0.35[a] | 20.75 ± 0.07[b] | 19.45 ± 0.21[c] |
| Lipids (%) | 19.72 ± 1.94[a] | 18.61 ± 3.25[a] | 19.15 ± 0.49[a] | 22.50 ± 0.56[a] |
| Ashes (%) | 2.50 ± 0.02[a] | 2.07 ± 0.01[b] | 1.95 ± 0.00[c] | 1.72 ± 0.03[d] |
| Proteins (%) | 6.86 ± 0.0[a] | 6.45 ± 0.08[b] | 4.14 ± 0.01[c] | 3.17 ± 0.02[d] |
| Carbohydrates (%) | 44.88 ± 1.62[a] | 46.60 ± 2.99[ac] | 54.01 ± 0.41[b] | 53.15 ± 0.41[bc] |
| Energy value (kcal) | 382.86 ± 10.90[a] | 379.73 ± 17.62[a] | 404.95 ± 2.76[ab] | 427.80 ± 3.53[b] |
| pH | 8.19 ± 0.01[a] | 6.60 ± 0.00[b] | 6.50 ± 0.00[c] | 6.30 ± 0.00[d] |

percentage of biosurfactant in cupcakes due to the lipid nature of glycolipids. It is important to note that about 75% of the fatty acids of *C. bombicola* glycolipid are unsaturated (C18:1, C18:2 and C18:3), while the other 25% are saturated fatty acids (mainly C16:0), unlike vegetable fat, which may contain trans fatty acids, which increase cholesterol levels and cause cardiovascular disease (*Cavendish et al., 2010*). Thus, it can be suggested that the replacement of vegetable fat by biosurfactant allows an improvement in the quality of the formulated dessert.

Regarding the ash content, all formulations produced presented these different levels, being observed a decline in their content during the substitution of vegetable fat by biosurfactant. The addition of the biosurfactant also acted significantly in the reduction of the fixed mineral residue, since the commercialized industrial vegetable fat used in the formulations had about 550 mg/100 g of product, representing a great contribution in the percentage of ash in the samples. Likewise, the protein content decreased as the incorporation of the biosurfactant increased.

The opposite effect was observed in relation to carbohydrates, that is, the increase in the concentration of biosurfactant from 75% favored the increase in the carbohydrate content in cupcakes. Since the biosurfactant is a glycolipid, it may have contributed to the increase in the carbohydrate content of the formulated product.

The energy value did not vary significantly between the standard formulation and formulations A and B, and there was no difference between formulations B and C. With the substitution of vegetable fat for biosurfactant, from 75%, the carbohydrate content was higher, implying higher energy values. *Zahn, Pepke & Rohm (2010)* reported the replacement of fat in muffins by carbohydrates and obtained a reduction of about 45% in the content of lipids and, consequently, a reduction in the caloric value of the product.

Regarding the pH of the cupcakes, there was a decrease when the biosurfactant was added to the formulation in increasing concentrations, but this decrease did not affect the quality of the cupcakes. Reducing the pH can be considered an advantage, since more basic

foods are more susceptible to microbial multiplication and, consequently, to deterioration (*Jay, Loessner & Golden, 2005*).

## DISCUSSION

Analyzing the surface and interfacial tensions obtained in this work for the *C. bombicola* biosurfactant with the values presented by *Jadhav, Pratap & Kale (2019)*, using *Starmerella bombicola* MTCC 1910 in medium containing 10% sunflower oil residue, similar results can be observed, from 35.5 ± 0.52 mN/m and 0.923 ± 0.06 mN/m for surface and interfacial tensions, respectively. Similar results were also found by *Shah et al. (2017)* in a medium with 10% palm oil, using *Starmerella bombicola* ATCC 22214, which obtained 35.35 mN/m for surface tension and 3.322 ± 0.176 mN/m for interfacial tension. Values of 35.33 ± 0.19 mN/m and 2.53 ± 0.02 mN/m for surface and interfacial tensions, respectively, were found by *Ribeiro et al. (2019)* using the yeast *C. utilis* UFPEDA 1009 cultivated in a low-cost medium supplemented with 6% residual canola oil, while *Elshafie et al. (2015)* using *C. bombicola* ATCC 22214 cultivated in glucose and corn oil observed that the biosurfactant produced reduced the both surface tension and interfacial tension to 28.56 ± 0.42 mN/m and 2.13 ± 0.09 mN/m, respectively, within 72 h. The sophorolipids produced by *Starmerella bombicola* cultivated in waste cooking oil reduced the surface tension to 32.6 mN/m and interfacial tension was 1.4 mN/m (*Maddikeri, Gogate & Pandit, 2015*). The yeast *Sporisorium* sp. aff. *sorghi* SAM20 produced approximately 32 g/L glycolipid biosurfactants from soybean oil after 7 days. The critical micelle concentration and the surface tension at CMC were estimated to be 20 mg/L and 30.0 mN/m, respectively (*Alimadadi, Soudi & Talebpour, 2018*).

Bacterial surfactants, on the other hand, such as rhamnolipids and surfactin, are capable to reduce the surface tension up to 25–26 mN/m, and the interfacial tension to values as low as 0.001 mN/m. Furthermore, they exhibit critical micelle concentrations considerably lower (10 mg/L) when compared with yeasts biosurfactants (*Henkel et al., 2017*; *Li, 2017*; *Tan & Li, 2018*).

Regarding the isolation of the biosurfactant, *Ribeiro et al. (2019)* used the same extraction methodology for the isolation of biosurfactant produced by *C. utilis* UFPEDA 1009 and obtained 24.22 ± 0.23 g/L. *Pinto et al. (2018)* obtained a yield of 61 g/L using *C. bombicola* in a bioreactor, with the same extraction methodology used in this work, indicating the feasibility of this method to increase the yield of production in biosurfactant. *Daverey & Pakshirajan (2010)*, on the other hand, for *C. bombicola* cultivated in a medium containing mixed hydrophilic substrate (deproteinized whey and glucose), yeast extract and oleic acid, obtained 23.29 ± 0.54 g/L, 25.54 ± 1.01 g/L and 33.32 ± 0.83 g/L of biosurfactant when fermentation was carried out in batch shake flasks, in bioreactor without pH control and in bioreactor with pH control, respectively. The use of waste cooking oil as substrate to produce biosurfactants by *Starmerella bombicola* in a batch fermentation in the presence of ultrasound gave 24.7 g/L of sophorolipids. The fed-batch mode of fermentation, on the other hand, gave 55.6 g/L of sophorolipids (*Maddikeri, Gogate & Pandit, 2015*). *Kaur et al. (2019)* described that a high sophorolipid process efficiency was achieved by fed-batch fermentation using restaurant food waste

hydrolysate as the batch medium. A sophorolipids titer of 115.2 g/L was obtained in a fermentation time of 92 h. *Shah et al. (2017)* used crude oils (Tapis oil, Melita oil and Ratawi oil) as substrates for biosurfactant production by *Starmerella bombicola*. The sophorolipids yields using Tapis, Melita, and Ratawi oil were 26 g/L, 21 g/L and 19 g/L, respectively. The sophorolipids reduced the surface tension of pure water to 36.38 mN/m, 37.84 mN/m and 38.92 mN/m, respectively, corresponding to critical micelle concentrations of 54.39 mg/L, 55.68 mg/L and 58.34 mg/L.

The emulsification results against vegetable oils obtained in our work can be compared to the values found in the literature for emulsification indexes of biosurfactants from *Candida* species. *Pinto et al. (2018)*, using *C. bombicola* to produce a biosurfactant from industrial waste obtained emulsification rates of 28% for canola oil, 26% for corn oil and 30% for soybean oil. *Gaur et al. (2019)* obtained 50%, 50%, 51% and 49% emulsification for almond, mustard, soy and olive oil, respectively, for the *C. glabrata* biosurfactant CBS138. *Campos, Stamford & Sarubbo (2014)*, for the biosurfactant produced by *C. utilis* UFPEDA 1009 in medium containing residual canola oil, obtained 73%, 73%, 43% and 33% of emulsification for sunflower, corn, soy and rice oils, respectively. The yeast *C. bombicola* ATCC-22214 cultivated in a medium supplemented with coconut oil and glucose under fed-batch culture conditions produced 54.0 g/L sophorolipids with significant surface activity (25–35 mN/m), and emulsion ability (*Morya et al., 2013*).

In the literature, it is not easy to find reports of thermostability in biosurfactants produced by *C. bombicola*; however, biosurfactants produced by species of bacteria have been characterized in terms of their thermostable profile. *Kiran et al. (2017)* analyzed the thermostability of the lipopeptide produced by *Nesterenkonia* sp and obtained similar results from melting peaks of 240 °C and 320 °C. The maximum loss of mass occurred at a temperature of 260 °C, with 50.725%. *Singh & Tiwary (2016)* produced a biosurfactant from *Pseudomonas otitidis* P4 and observed that at a temperature of 149.9 °C, 21.38% of the biosurfactant had been degraded. The crystallization temperature was similar to the temperature found for the *C. bombicola* biosurfactant, that is, 112.83 °C.

According to *Han et al. (2015)*, the higher the melting peak of the biosurfactants, the greater is their thermal stability. In addition, compounds that exhibit high degradation temperatures are more advantageous, since they are more resistant to extreme temperature conditions. Thus, knowing that the cupcakes cooking process takes place at a maximum temperature of 180 °C, it is possible to state that the biosurfactant will not suffer significant loss of mass, which makes its application feasible.

The literature does not describe the antioxidant activity with biosurfactants produced by the yeast *C. bombicola*. However, lipopeptides have been described as excellent antioxidants, mainly in relation to the elimination of the DPPH radical (*Tabbene et al., 2012*; *Yalçin & Çavuşoğlu, 2010*). *Ben Ayed et al. (2015)* reported an interesting activity of eliminating DPPH radicals (65%) of lipopeptides from *Bacillus mojavensis* A21. Another biosurfactant produced by *Lactobacillus* species showed sequestering activity of DPPH radicals around 75% at a concentration of 5,000 μg/mL (*Merghni et al., 2017*). The biosurfactant from *Streptomyces* sp. showed a total antioxidant capacity of 80.4% at a

concentration of 200 µg/mL (*Ramrajan et al., 2017*), while the mannosylerythritol lipids produced by *Pseudozyma hubeiensis* showed 50% DPPH radical scavenging activity at 10 mg/mL and sequestration of superoxide ions greater than 50% under a concentration of one mg/mL (*Takahashi et al., 2012*). Thus, the biosurfactant produced by *C. bombicola*, which showed 52.42% activity at a concentration of 1,000 µg/mL, can be considered a good antioxidant, according to the DPPH radical scavenging method.

In the literature, it is common to find reports of bacterial biosurfactants that are nontoxic. For comparative purposes, the biosurfactant used in this work is derived from a yeast; however, it is important to mention some reports that may corroborate with the results found for the *C. bombicola* biosurfactant, since it is a glycolipid class biosurfactant, as described by *Luna et al. (2016)*. *Patowary et al. (2017)* used a rhamnolipid produced by *P. aeruginosa* PG1 and obtained 85.6% cell viability for the mouse L292 fibroblast cell line. Another glycolipidic surfactant produced by *Rhodococcus* sp. 51T7 showed less toxicity than synthetic surfactants against 3T6 mouse fibroblasts (*Marques et al., 2009*).

## CONCLUSIONS

The partial or total replacement of vegetable fat by the biosurfactant produced by *C. bombicola* URM 3718 did not drastically affect the final product, indicating the feasibility of applying this biomolecule in cupcakes formulation. The biosurfactant still has thermal stability, antioxidant properties and is nontoxic, being a potential additive for the food sector. The replacement of vegetable fat for biosurfactant can also improve the nutritional value of flour meal by reducing the trans fatty acids present in vegetable fat. The results obtained in this research demonstrate the feasibility of future studies focused on scale-up production of the biosurfactant, as well as conducting sensory studies aiming its future and promising application as an emulsifying additive for the food industry.

## ACKNOWLEDGEMENTS

Authors like to thanks Lucas Rocha for his technical assistance.

### Funding
This work was supported by the following Brazilian fostering agencies: State of Pernambuco Assistance to Science and Technology Foundation (FACEPE), National Council for Scientific and Technological Development (CNPq), Coordination for the Advancement of Higher Education Personnel (CAPES) and the National Electrical Energy Agency (ANEEL). The funders had no role in study design, data collection and analysis, decision to publish, or preparation of the manuscript.

### Grant Disclosures
The following grant information was disclosed by the authors:
State of Pernambuco Assistance to Science and Technology Foundation (FACEPE).

National Council for Scientific and Technological Development (CNPq).
Coordination for the Advancement of Higher Education Personnel (CAPES).
National Electrical Energy Agency (ANEEL).

## Competing Interests

The authors declare that they have no competing interests.

## Author Contributions

- Ivison A. Silva performed the experiments, analyzed the data, prepared figures and/or tables, and approved the final draft.
- Bruno O. Veras performed the experiments, prepared figures and/or tables, and approved the final draft.
- Beatriz G. Ribeiro performed the experiments, prepared figures and/or tables, and approved the final draft.
- Jaciana S. Aguiar performed the experiments, prepared figures and/or tables, and approved the final draft.
- Jenyffer M. Campos Guerra performed the experiments, analyzed the data, prepared figures and/or tables, authored or reviewed drafts of the paper, and approved the final draft.
- Juliana M. Luna performed the experiments, analyzed the data, prepared figures and/or tables, and approved the final draft.
- Leonie A. Sarubbo conceived and designed the experiments, analyzed the data, prepared figures and/or tables, authored or reviewed drafts of the paper, and approved the final draft.

## Data Availability

Data are available as Supplemental Files.

## Supplemental Information

Supplemental information for this article can be found online at http://dx.doi.org/10.7717/peerj.9064#supplemental-information.

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
