# Peer review of "Production of cupcake-like dessert containing microbial biosurfactant as an emulsifier"

_PeerJ, doi:10.7717/peerj.9064_

## Round 0.1 · original submission · Major Revisions

· Academic Editor

Major Revisions

It is an interesting (possible) application of glycolipids type biosurfatcant in food industry. The methods reported are standard and conclusive. It needs some clarifications, few typos needs to be corrected, and some additional information. Such as:
- Please correct references (in-text) as per AI - for eg:. (Bandyopadhyay, Chakraborty & Bhattacharyya, 2014); (Campos, Stamford & Sarubbo, 2014) - and throughout manuscript.
- 'The yeast inoculum was standardized by transferring a young culture to flasks containing 50 mL of YMB medium,,' - How? How much WAS transferred?
- Critical Micelle Concentration (CMC) - L107-109 Methods described are not clear.
- Many references are quite old: Re et al. (1999); (Blois, 1958); Alley et al. (1988) and Mosmann (1983) - If possible please cite some recent references for such methods.
- 'For the calibration curve, a standard Trolox solution was prepared, at a concentration of 10 to 200 μM, as previously described.' - Where is it described?
- 'The percentage inhibition was calculated using GraphPad Prism 7.0 demo software. The percentage inhibition was calculated using GraphPad Prism 7.0 demo software.' - Sentences are repeated.
- 'To determine the energy value, the mass values of carbohydrates, lipids and proteins were multiplied by 4, 9 and 4, respectively, and the values were added later.' - Why multiplied by those numbers/Factors? Please explain.
- 'The biosurfactant yield obtained was 25 g/L.' - +/- SD?
- Although not specifically required, To develop acceptable food products it is necessary to measure the Salivary and sensory perceptions and hedonic responses of consumers. How does your cup cakes taste? Was there any difference?
- 'In the literature, it is not easy to find reports of thermostability in biosurfactants produced by C. bombicola' - Well not correct, please refer one mentioend below and similar others by this group and others:
Elshafie, A. E., Joshi, S. J., Al-Wahaibi, Y. M., Al-Bemani, A. S., Al-Bahry, S. N., Al-Maqbali, D., & Banat, I. M. (2015). Sophorolipids production by Candida bombicola ATCC 22214 and its potential application in microbial enhanced oil recovery. Frontiers in microbiology, 6, 1324.
- What about economics (while replacing vegetable oils by BS)?
- 'biomolecule in cookies formulation.' - Cup cakes.
- Captions for some of the tables (6, 7) and figures (2) are incomplete. Please check and correct, if not due to formatting and uploading issue.

For more details please also refer to the reviewers comments.

Reviewer 1 ·

Basic reporting

The manuscript is in general well written. The authors use numerous references from their group (around 20%) to discuss the results. In order to put their data into context, a more diverse literature should be used. Tables and figures are appropriate.

Experimental design

The subject of the work is within the scope of the journal and is relevant. The methodologies used are well described.

Validity of the findings

The alternative formulation suggested by the authors does not provide clear advantages when compared with the original formulation, and it is more expensive.

Additional comments

The manuscript “Production of cupcake-like dessert containing microbial biosurfactants as an emulsifier” studies the application of the biosurfactant produced by Candida bombicola URM3718 in the elaboration of cupcakes. The biosurfactant was produced using a residue-based culture medium, which resulted in the production of 25 g of biosurfactant per liter. The interfacial activity of the biosurfactant was studied through surface/interfacial tension measurements and through microscopy. The thermal stability of the biosurfactant was also studied through differential scanning calorimetry and thermogravimetric analysis. The biosurfactant was used to replace vegetable fat in cupcakes’ formulation. According to the authors, “cupcakes with biosurfactant incorporated in their dough did not show significant differences in physical and physical-chemical properties after baking when compared to the standard formulation”. Biosurfactants are outstanding molecules with remarkable properties. Their application in the food industry to replace vegetable fat in dough formulation, as suggested in this work, is out of consideration. The amount of biosurfactant used in this work (between 80 and 160 g per Kg of dough) is very high. It is expected the use of biosurfactants, in low amounts, for instance to replace conventional synthetic emulsifiers, and it is expected that their incorporation in dough formulation will improve the properties of the final product. Otherwise, the application of biosurfactants in the food industry will not be economically viable, even if they are produced using low-cost culture media.
Lines 257-261. “The results obtained for surface and interfacial tension for the biosurfactant produced by C. bombicola were 30.790 ± 0.04 mN/m and 0.730 ± 0.05 mN/m, respectively. According to Akbari et al. (2018), biosurfactants with the ability to reduce the surface tension of water from 72 to 35 mN/m and the interfacial tension from 40 to 1 mN/m are effective, being considered good surfactants” However, other biosurfactants such as rhamnolipids and surfactin (and other lipopeptides) reduce the surface tension up to 25-26 mN/m, and the interfacial tension to values as low as 0.001 mN/m. Furthermore, they exhibit critical micelle concentrations considerably lower (10 mg/L) when compared with the biosurfactant produced by C. bombicola URM3718. The authors should compare the biosurfactant herein studied not only with biosurfactants previously reported that exhibit a similar or worst performance, but also with biosurfactants that exhibit better properties. In that sense, more appropriate references should be used. In summary, this biosurfactant exhibits a weak activity when compared with others previously reported.
Lines 264-268. An advantageous method was used to recover the biosurfactant, avoiding centrifugation and filtration (i.e. without removing the cells). How does this affect to the purity of the biosurfactant recovered? Which was the purity of the biosurfactant used in this work?
Lines 273-276. “CMC is related to the efficiency of surfactants and ranges from 0.001 to 2 g/L for biosurfactants. Biosurfactants are more efficient as they have reduced CMC (Santos, Luna, Rufino, Santos & Sarubbo, 2016). Thus, the C. bombicola biosurfactant proved to be efficient, since it had a CMC of 0.5 g/L”. Once again, the biosurfactant herein studied is considerably less efficient when compared with those that exhibit considerably lower CMCs. That point should be discussed in the manuscript. By the way, in the Abstract it is referred that the CMC is 0.6 g/L.
Regarding the emulsifying activity, the authors state that “the increase in the concentration of biosurfactant did not exert a significant increase over the emulsification percentages”, although biosurfactant concentrations below and above the CMC were studied. Biosurfactants usually exhibit higher emulsifying activities at concentrations above the CMC, how can the authors explain this behavior for this biosurfactant?
The results obtained in the study of the antioxidant activity of this biosurfactant should be compared with those obtained for other biosurfactants (or similar compounds) in order to assess the potential of this biosurfacant.
Regarding the characterization of the cupcakes elaborated with the different formulations, “The results demonstrate that there were no major discrepancies between the different formulations in relation to the evaluated parameters… the partial or total replacement of vegetable fat by the biosurfactant did not cause significant changes in the physical properties of the cupcakes”. Which is the advantage of replacing vegetable fat by a more expensive ingredient (biosurfactant)?
Lines 382-384: “As a consequence, cupcakes containing vegetable fat in their composition had a higher moisture content compared to cupcakes that contained biosurfactant in the formulation”. This seems to be a disadvantage.
Lines 386-387: “Considering that the C. bombicola biosurfactant has a glycolipidic nature (Luna et al., 2016)”. Lines 389-391: “about 75% of the fatty acids of C. bombicola glycolipid are unsaturated (C18: 1, C18: 2 and C18: 3), while the other 25% are saturated fatty acids (mainly C16: 0)” This characterization corresponds to the same biosurfactant herein studied? Which is the exact chemical composition of this biosurfactant, including the carbohydrate fraction?
Lines 403-405: “Since the biosurfactant is a glycolipid formed by carbohydrate units called glycans, it may have contributed to the increase in the carbohydrate content of the formulated product.” This is a general assessment. Which is the structure of the glycidic fraction of the biosurfactant herein studied?
Lines 412-415: “Regarding the pH of the cupcakes, there was a decrease when the biosurfactant was added to the formulation in increasing concentrations. This occurred due to the pH value of the biosurfactant production medium, which at the end of the fermentation reaches values around 6.2” The biosurfactant was not purified from the culture medium before its addition to the dough? Is it not possible to adjust the pH of the biosurfactant solution to avoid these variations? I cannot understand which is the effect of the pH of the culture medium in the pH of the final formulation.
Lines 418-432: The authors compare the surface activity and the amount of biosurfactant produced by the yeast strain herein studied with the data obtained by other researchers. However, most of these works come from the same group! The authors should take a look to other works from other researchers reporting biosurfactants produced by different yeasts with better surface-active properties. The same can be said regarding the emulsifying activity (lines 433-441).
A.E. Elshafie, S.J. Joshi, Y.M. Al-Wahaibi, A.S. Al-Bemani, S.N. Al-Bahry, D.Al-Maqbali, I.M. Banat, Sophorolipids production by Candida bombicola ATCC22214 and its potential application in microbial enhanced oil recovery, Front. Microbiol. 6 (2015) 1324
G.L. Maddikeri, P.R. Gogate, A.B. Pandit, Improved synthesis of sophorolipidsfrom waste cooking oil using fed batch approach in the presence ofultrasound, Chem. Eng. J. 263 (2015) 479–487.
Kaur G et al. 2019. Efficient sophorolipids production using food waste. Journal of Cleaner Production 232:1-11.
Alimadadi N et al. 2018. Efficient production of tri-acetylated nono-acylated mannosylerythritol lipids by Sporisorium sp. Aff. Sorghi SAM20. J. Appl. Microbiol. 124(2):457-468.
Lines 430-432: “Pinto et al. (2018) obtained a yield of 61 g/L using C. bombicola in a bioreactor, with the same extraction methodology used in this work, indicating the feasibility of this method to increase the yield of production in biosurfactant” I do not believe that the extraction method used has influence in the amount of biosurfactant produced.

Reviewer 2 ·

Basic reporting

The manuscript number 46042 described biosurfactant production by Candida bombicola URM 3718 and its possible application in food industry.

The work is clear presented and included relevant results

Experimental design

no comment

Validity of the findings

no comment

Additional comments

1. The authors do not describe about the purification rate of the biosurfactant, could the authors specify if the crude or the purified biosurfactant was used in this study.
2. Nature of the biosurfactant was not established in this study, could the authors explain why they refer to a previous study of biosurfactant production by the same strain even the extraction method was different and maybe some characteristic of the biosurfactant too?
3. The authors used the fact that the biosurfactant was a glycolipid in nature to explain the decrease of protein and the increase of the carbohydrate in the cupcakes when the concentration of the biosurfactant was increased, while the composition of the biosurfacant given in the reference was “The preliminary analysis demonstrated that the biosurfactant isolated from C. bombicola consisting of 70% lipids, 10% carbohydrates and 20% proteins”. (Luna et al, 2016). Could the authors explain this?
4. Could the authors discuss if the decrease of pH from 8.19 to 6.3-6.6 can affect the quality of the cupcakes?
5. In the conclusion, line 478: cookies or cupcakes?

·

Basic reporting

Paper is very much clear. Professional English is used
Recent references are also cited well

Experimental design

Experimental design is also fine

Validity of the findings

Need to improve information in the table and figure with respective to concentrations. Figures are not labelled properly

Additional comments

Isolation of biosurfactant
99 A method developed in the laboratory was used, which initially consisted of extraction with ethyl100 acetate, twice, in the proportion 1:4 (v/v) of the non-centrifuged broth. Then, the organic phase 101 was centrifuged (2600xg for 20 minutes) and filtered. The filtrate was transferred back to the 102 separating funnel and a saturated NaCl solution was added to separate the remaining aqueous 103 phase. The organic phase was transferred to an Erlenmeyer flask and anhydrous MgSO4 was 104 added until granules were formed, which were filtered and dried at 50 ° C.

What about other debris and metabolites? It is not mentioned anywhere how purified biosurfactant was purified?

Table 5(on next page)
Percentage of viability of the C. bombicola URM 3718 biosurfactant against the L929 and the Vero strains.
What is the concentration? It is not mentioned in the table.

Physical properties of cupcakes after baking for the standard formulation (100% vegetable fat), formulation A (50% vegetable fat + 50% biosurfactant), formulation B (25% vegetable fat + 75% biosurfactant) and formulation C (100% biosurfactant). Data expre…….
Typo errors
At many places such type errors are seen.
For example
Physical-chemical composition and energy value of cupcakes for the standard formulation (100% vegetable fat), formulation A (50% vegetable fat + 50% biosurfactant), formulation B (25% vegetable fat + 75% biosurfactant) and formulation C (100 % biosurfacta

Physical properties of cupcakes after baking for the standard formulation (100%
vegetable fat), formulation A (50% vegetable fat + 50% biosurfactant), formulation B
(25% vegetable fat + 75% biosurfactant) and formulation C (100% biosurfactant). Data
expre

Table 4(on next page)
. Percentage of sequestration of the radical 2,2’-azino-bis (3-ethylbenzothiazoline-6-
sulfonate) (ABTS•+) by the biosurfactant. Data expressed as mean ± S.D. of triplicate
determinations.

Table 5(on next page)
Percentage of viability of the C. bombicola URM 3718 biosurfactant against the L929 and the Vero strains.
What is the concentration? It is not mentioned in the table.

Figure 1
Critical Micelle Concentration (CMC) of the biosurfactant isolated from Candida bombicola URM 3718
Indicate CMC value in the figure
Information can be represented with CMC concentration and suface tension value

---

## Round 0.2 · accepted · Accept

· Academic Editor

Accept

Authors incorporated the majority of suggestions and amended the manuscript considering all comments.

Reviewer 1 ·

Basic reporting

The manuscript is in general well written. Tables and figures are appropriate.

Experimental design

The subject of the work is within the scope of the journal and is relevant. The methodologies used are well described.

Validity of the findings

The alternative formulation suggested by the authors does not provide clear advantages when compared with the original formulation, and it is more expensive.

Additional comments

The answers given by the authors to my questions are not convincing to change my evaluation. Some questions were not properly answered, as for example those related with the purity and the chemical structure of the biosurfactant herein used. The main concern is that the alternative dough formulation suggested by the authors did not provide clear advantages when compared with the original formulation, and it is considerably more expensive.
According to the answers given by the authors, “this study presents an initial study aiming to evaluate if the biosurfactant (at high concentrations) can exert the same effect as the vegetable fat in physical, physicochemical parameters and texture profile of cupcakes”. The final result is a more expensive formulation (the cost of the biosurfactant is considerably higher when compared to the vegetable fat) which does not improve the properties of the final product.
Also, according to the authors, “the results obtained serve to support future studies that seek to decrease these (biosurfactant) concentrations and increase industrial viability. Studies with lower (biosurfactant) concentrations, based on the positive results obtained in this study, together with the sensory evaluation, are going on” Maybe the authors should focus in those studies, using lower biosurfactant concentrations, in order to obtain more promising results.
Although I agree with the statement of the authors: “there is an increase interest of the market for products of better quality”, it is also true that consumers also look at the price. Which was the real improvement of the properties of cupcakes containing biosurfactant when compared with those containing vegetable fat? And which would be the increase of production costs?

Reviewer 2 ·

Basic reporting

The manuscript number 46042:1 described biosurfactant production by Candida bombicola URM 3718 and its possible application in food industry.


New references are added to the revised version of this paper

Experimental design

The experimental design is well done

Validity of the findings

The work is clear presented and included relevant results.

·

Basic reporting

Authors have addressed all the queries

Experimental design

Revised manuscript is satisfactory

Validity of the findings

They are well supported

Additional comments

Authors have addressed all the queries satisfactorily